# Early Use of Innovative Biomarkers Such as Mid-Regional Pro-Adrenomedullin and SeptiCyte^®^ RAPID in Post-Cardiac Surgery Patients: Pilot Case Series

**DOI:** 10.3390/ijms26199453

**Published:** 2025-09-27

**Authors:** Chiara Risso, Lorenzo Vay, Francesca Sciascia, Riccardo Traversi, Marco Ellena, Anna Chiara Trompeo, Luca Brazzi, Giorgia Montrucchio

**Affiliations:** 1Department of Surgical Sciences, University of Turin, 10126 Turin, Italy; lorenzo.vay@unito.it (L.V.); francesca.sciascia@unito.it (F.S.); riccardo.traversi@unito.it (R.T.); luca.brazzi@unito.it (L.B.); 2Anestesia e Rianimazione 1 U, Department of Anesthesia, Intensive Care and Emergency, Città della Salute e della Scienza Hospital, 10126 Turin, Italy; mellena@cittadellasalute.to.it (M.E.); atrompeo@cittadellasalute.to.it (A.C.T.)

**Keywords:** midregional pro-adrenomedullin, SeptiCyte RAPID, cardiopulmonary bypass, inflammatory response, endothelial dysfunction, host gene response, intensive care unit, biomarkers, host immune response, endocarditis

## Abstract

Prognostic uncertainty and missed diagnoses of sepsis remain frequent after cardiopulmonary bypass (CPB) surgery, where systemic inflammatory response (SIRS) arises from surgical trauma, blood activation in the extracorporeal circuit, ischemia/reperfusion injury, and endotoxin release. Among innovative biomarkers, pro-adrenomedullin (pro-ADM), particularly its stable fragment mid-regional pro-adrenomedullin (MR-proADM), has shown promise for detecting endothelial dysfunction and predicting organ failure in sepsis. SeptiCyte^®^ RAPID (Seattle, WA, USA) also represents a novel diagnostic tool that assesses the host immune response by quantifying PLA2G7 and PLAC8 gene expression in whole blood, offering potential for early differentiation between sepsis and sterile inflammation. We analyzed traditional and innovative biomarkers within 24 h post-CPB in a pilot group of patients admitted to the cardiac Intensive Care Unit of the “Città della Salute e della Scienza” University Hospital (Turin, Italy) between June and November 2023. Data from the following 14 patients were collected: 7 undergoing surgery for infective endocarditis (IE, Group 1) and 7 having standard elective cardiac surgery (Group 2). Procalcitonin (PCT), lactate, and pro-ADM increased in Group 1 but not in Group 2. SeptiCyte^®^ RAPID showed a moderate, borderline increase in Group 1. The innovative biomarkers had a good performance in patients exhibiting signs of organ dysfunction and in subjects demonstrating at least cardiovascular and/or pulmonary damage and under vasopressor and inotropic support. Although limited by the small sample, our preliminary data suggest no biomarker alterations in patients with standard elective cardiac surgery, unlike in those with IE.

## 1. Introduction

Sepsis is a potentially lethal medical condition characterized by a dysregulated reaction of the host to an infecting pathogen. Early recognition, hemodynamic restoration, and antimicrobial administration represent the foundation of “sepsis bundles” [1].

However, unlike for other entities, no fast gold standard univocal diagnostic strategy has been found. Uncertainty and missed diagnoses of sepsis are still frequent, especially early in the course of illness when no site of infection or pathogen has been identified and when organ damage has not yet become evident. There is, thus, an urgent and unsatisfied need for new diagnostics, mainly for new biomarkers, available in a timely and cost-effective manner [2].

Since many of the cellular pathways that are activated in response to infections are also activated in response to tissue trauma and non-infectious inflammation, the real challenge is to find a way to differentiate systemic inflammatory response syndrome (SIRS), which is an excessive defensive body response to a harmful stressor (trauma, surgery, acute inflammation, ischemia or reperfusion, cancer), from infection-triggered organ dysfunction. In the intensive care setting, differentiating between the various types of shock represents an additional challenge—particularly between septic shock and cardiogenic shock, or, in cases of mixed shock, in determining the extent to which a septic component is present.

Recently, increasing attention has been devoted to the so-called immunological biomarkers, which are capable not only of characterizing the inflammatory component of tissue injury, but also of identifying the potential role of the immune system as a marker of a specific response [3]. New panels of biomarkers have been proposed for interrogating the host immune response.

It is specifically in this context that SeptiCyte^®^ LAB and SeptiCyte^®^ RAPID emerge, two validated devices that measure the expression of genes which are indicative of a dysregulated immune response during sepsis [4].

SeptiCyte^®^ RAPID, in particular, evaluates the expression of the genes PLA2G7 and plac8 in whole blood by quantitative reverse transcriptase polymerase chain reaction (RT-qPCR). PLA2G7 encodes the platelet-activating factor (PAF) protein acetylhydrolase, an enzyme produced mostly by macrophages whose action leads to the release of pro-inflammatory mediators (lysophospholipids and oxidized fatty acids). High plasma levels have been found to correlate with sepsis survival and reduced levels have been found in sepsis. Plac8 is an interferon-inducible gene and is expressed on a wide variety of immune cells (spleen, lymph nodes), including plasmacytoid dendritic cells. In sepsis, it is upregulated in different types of peripheral blood cells. SeptiCyte generates a quantitative score (SeptiScore) that increases with an increasing likelihood of sepsis.

Recent publications investigated the reliability of the assay and it turned out to be a promising diagnostic tool in the assessment of bacterial, viral, and fungal infection likelihood in critically ill adult patients with systemic inflammation, even if immunocompromised, particularly when combined with clinical assessment and laboratory variables.

Supporting this evidence, Balk et al. (2024) demonstrated that SeptiCyte^®^ RAPID maintained a high diagnostic accuracy in distinguishing between sepsis and SIRS across different ICU patient subgroups, including those with varying comorbidities, treatments, infection types, and demographics [5]. Its performance was not significantly affected by antibiotic timing or the use of vasopressors or immunosuppressors.

Furthermore, a 2023 French study confirmed the potential utility of SeptiCyte^®^ RAPID in the risk stratification of COVID-19 patients, based on clinical severity evaluated through chest CT imaging and/or ICU admission [6].

Diagnosing infections and sepsis can be especially challenging in perioperative settings, even for experienced clinicians. Traditional host biomarkers like C-reactive Protein (CRP) and procalcitonin (PCT) have limited utility in the context of surgery-induced inflammatory responses, drug- or surgery-induced immunodeficiency, underlying malignancies, and high levels of comorbidity.

In particular, distinguishing between SIRS, cardiogenic shock, and septic shock in post-cardiac surgery patients remains a clinical challenge due to the complex pathophysiological response triggered by intraoperative cardiopulmonary bypass (CPB) and cardioplegia. In this context, it is essential for clinicians to rapidly identify cases where an infectious component is involved and to initiate timely and appropriate medical management, such as the prompt administration of antibiotics.

A promising inflammatory biomarker potentially useful in the early recognition of endothelial damage and commonly reported in cases of septic shock and infection is adrenomedullin (ADM), a multipotent regulatory peptide with several biological activities—vasodilator, positive inotropic, diuretic, natriuretic, and bronchodilator—widely expressed throughout the body and produced by multiple tissues in response to an infective stimulus. Due to its increased stability, the precursor of ADM, mid-regional proADM (MR-proADM), is clinically measured instead.

Previous studies have identified MR-proADM as a potential candidate biomarker to predict mortality and treatment response in septic patients [7] and severe COVID-19 [8]. Less is known about the potential role of MR-proADM in post-cardiac surgical patients, but a previous study explored the course of adrenomedullin and endothelin-1 levels in patients with vasodilatory shock after cardiac surgery. Significantly higher levels of both biomarkers were associated with organ dysfunction, and different courses of both biomarkers were observed in patients with vasodilatory shock after cardiac surgery [9]. In addition, a post hoc analysis of the HERACLES randomized controlled trial evaluated the predictive value of MR-proADM in the fluid overload in post-cardiac surgery, finding no statistically significant association between MR-proADM and fluid overload at ICU discharge or day 6 post-surgery—although elevated levels were observed in patients with fluid overload [10].

The aim of our preliminary study is to evaluate the diagnostic performance of the molecular host response assay SeptiCyte^®^ RAPID and the endothelial damage biomarker MR-proADM in critically ill adults post-CPB, in order to provide clinical evidence regarding the reliability of these tests also in the postoperative Intensive Care Unit (ICU) setting of cardiac surgery.

For this purpose, we decided to assess the response of SeptiCyte^®^ RAPID and MR-proADM in the following two categories of cardiac surgery patients: those undergoing standard elective cardiac surgery and those undergoing valve replacement due to infective endocarditis with a surgical indication (the type of surgery is reported in Table 1 and Table 2 for both groups).

## 2. Results

Group 1—infective endocarditis (IE)—was composed of seven patients, with a medium age of 69 years and a few comorbidities, such as hypertension (HT), chronic obstructive pulmonary disease (COPD), and diabetes mellitus (DM). All of them were critically ill, with a median Sequential Organ Failure Assessment (SOFA) score of 11 and a median Simplified Acute Physiology Score (SAPS) of 57. In the immediate post-CPB period, there was a substantial need for hemodynamic support: five patients required adrenaline, six received noradrenaline, and four of them needed both agents.

Group 2—elective cardiac surgery—was composed of seven patients, with a medium age of 61 years, fewer comorbidities, a SOFA and SAPS of 5 and 24, respectively, and a moderate need for pharmacological circulatory support.

The median CBP time was 184 min for Group 1 and 162 min for Group 2.

The pathogens responsible for infective endocarditis were Methicillin-Sensitive *Streptococcus epidermidis* (3/7 cases) and Methicillin-Resistant *Streptococcus epidermidis*, *E. faecalis*, *E. coli*, and *Candida albicans* (one case each).

The mean ICU length of stay was 10 days (±8.5 days) for subjects in Group 1 and 3 days (±5.7 days) in Group 2. No patients died in the first 28 days after surgery.

Detailed characteristics of patients with infective endocarditis (Group 1) are presented in Table 1, while those of patients undergoing elective cardiac surgery (Group 2) are summarized in Table 2.

PCT and lactate were incremented in Group 1, but not in Group 2. The median MR-proADM in the infective endocarditis cohort was 5.9 ng/mL versus 1.1 ng/mL in the IE group and the elective cardiac surgery group, respectively.

SeptiCyte^®^ RAPID showed a moderate, borderline increase in Group 1 with a median score of 5,1. Overall, samples corresponded to probability band 2, which resulted in “sepsis-likely” label according to the manufacturer’s specification. No increase was observed in Group 2 (Table 3 and Table 4, Figure 1).

## 3. Discussion

Despite being preliminary and based on a very limited sample size, our data highlight the good performance of SeptiCyte^®^ RAPID in yielding negative results in patients undergoing CPB without risk of infectious complications and subsequent early development of septic shock. An elevation in score up to range 2 (moderate probability of sepsis) was instead observed in five out of seven patients from the group treated for infective endocarditis (IE), who were characterized by greater clinical severity and the need for inotropic/vasopressor support. The endothelial injury biomarker MR-proADM also showed higher levels in patients treated for IE compared to those undergoing standard elective cardiac surgery.

The inflammatory response that occurs during CPB seems to result from a complex interaction that triggers the activation of cellular and humoral inflammatory mediators, along with the involvement of fibrinolytic and hemostatic systems [11,12]. This combination of specific mechanisms plays a role in the process. The early phase is triggered by the immediate surgical trauma and by the contact of blood with the synthetic extracorporeal circuit, while the late phase, occurring after the release of the aortic cross-clamp, is driven by ischemia–reperfusion and endotoxemia. Hypothermia and transfusion are also two nonspecific factors activating the inflammatory response.

Patients undergoing extracorporeal circulation, therefore, represent an excellent and challenging population in which to examine the ability of new biomarkers such as MR-proADM and SeptiCyte^®^ RAPID to distinguish between sepsis and non-infectious systemic inflammation.

The literature currently reports only one study evaluating SeptiCyte in cardiac surgery, which compares its performance in children undergoing congenital cardiac defect corrective surgery requiring CPB and in children admitted with new-onset, community-acquired, definite or highly probable cases of bacterial sepsis [13]. This point of care test has been tested and validated in critically ill adults admitted to the ICU, particularly when affected by a respiratory infection [4], as well as in post-gastrointestinal surgery patients [14], and recent publications analyzed its fluctuation in COVID-19 patients [6,15]. These findings collectively highlight the versatility and clinical relevance of SeptiCyte^®^ RAPID as both a diagnostic and risk stratification tool across a broad spectrum of critical conditions; its accuracy, robustness, and rapid turnaround time suggest that SeptiCyte could offer significant clinical utility for both acute triage and longitudinal patient monitoring.

Our observations showed no important biomarker alterations in patients undergoing standard elective cardiac surgery, while they were both increased in patients with IE requiring CPB surgery for valvular replacement. The tests showed a good performance in patients exhibiting signs of organ dysfunction (SOFA and SAPS) and in subjects demonstrating at least a cardiovascular and/or pulmonary damage and under vasopressor and inotropic support. Consistent with these findings, one of our patients (Patient 7) from the elective cardiac surgery group also showed markedly elevated levels of both MR-proADM and SeptiCyte^®^ RAPID. This initially unexpected result was promptly clarified by the identification of a documented *P. aeruginosa* pneumonia, supporting the clinical suspicion of septic shock. At the time of enrollment, however, there were no clinical, biochemical, or radiographic signs suggestive of infection, and for this reason, the patient was retained in the original group, consistent with an intention-to-treat approach.

In particular, MR-proADM is a peptide directly measuring the blood levels of adrenomedullin, but biochemically more stable and easier to determine. It is a multipotent regulatory peptide expressed in different organs and tissues, an indicator of endothelial damage thanks to its role in vascular permeability, regulation of the inflammatory cascade, and microcirculation stability [16]. High levels of MR-proADM have been reported in patients with respiratory infections and sepsis [17,18]. It has been used as a marker to stratify disease severity and mortality risk and as a predictor of prognosis [19,20].

Plasma concentrations of MR-proADM have been shown to be slightly elevated compared to healthy controls in cardiac surgery patients [21]. An increase in MR-proADM was seen already preoperatively and reached statistical significance at weaning from CPB. The increase was theoretically expected as a response to vasodilatory and hypotensive effects during anesthesia, surgery, CPB, and volume overload, since the action of ADM is to reduce hyperpermeability during severe inflammatory states. Nevertheless, these levels, however, were lower than levels measured in patients suffering from sepsis.

## 4. Materials and Methods

We prospectively enrolled 14 patients admitted to the postoperative cardiac surgery ICU of AOU Città della Salute e della Scienza Hospital in Turin (Italy) between June and November 2023. The study protocol received approval from the local Ethics Committee (Study number 00548/2020). The data collection process was conducted anonymously, and the need for individual consent was waived by the Local Ethics Committee.

All subjects underwent cardiac surgery with cardiopulmonary bypass: Group 1 had surgery for IE and Group 2 was subjected to elective standard cardiac surgery. Infective endocarditis was defined according to the updated Duke criteria of 2023 [22].

Within the 24 h prior to surgery, each patient underwent a comprehensive clinical assessment, including laboratory tests and any additional investigations deemed appropriate by the attending anesthesiologist or cardiothoracic surgeons. Postoperatively, patients were managed in the cardiac intensive care unit with individualized monitoring, encompassing routine laboratory tests and microbiological analyses when indicated by the responsible clinicians, to identify or rule out potential sources of sepsis.

Multiple biomarkers were analyzed, including PCT, lactate, MR-proADM, and SeptiCyte score. Blood samples were collected within the first 24 h post-CPB.

SeptiCyte score was calculated through the SeptiCyte^®^ RAPID point of care test, which consists of nucleic acid extraction RT-qPCR. The results are reported in 1 h on the Idylla™ platform. This score ranges from 0 to 15 and can be classified into four probability bands according to the manufacturer’s guidelines. A SeptiScore of ≤3.0 (band 1) suggests that sepsis is unlikely, while scores of 3.1–4.4 (band 2), 4.5–5.9 (band 3), and ≥6 (band 4) indicate progressively higher probabilities of sepsis.

## 5. Conclusions

Although very limited in sample size, our findings show that SeptiCyte^®^ RAPID and MR-proADM levels do not significantly increase after CPB unless there is an underlying infectious substrate, such as IE, or in cases of sepsis or septic shock. In this sense, when differentiation between SIRS, cardiogenic shock, and septic shock is required, the combined use of novel biomarkers (SeptiCyte^®^ RAPID and MR-proADM) and classical markers (CRP and PCT), alongside clinical parameters and medical judgment, can support physicians in making more confident therapeutic decisions during the early postoperative period following cardiac surgery. These results emphasize how sepsis diagnosis requires exploring new patterns, including probing of the molecular response and degree of cell activation, as for SeptiCyte^®^ RAPID. Further data are needed to better define the performance of this new test in different clinical settings, such as post-cardiac surgery.

## Figures and Tables

**Figure 1 ijms-26-09453-f001:**
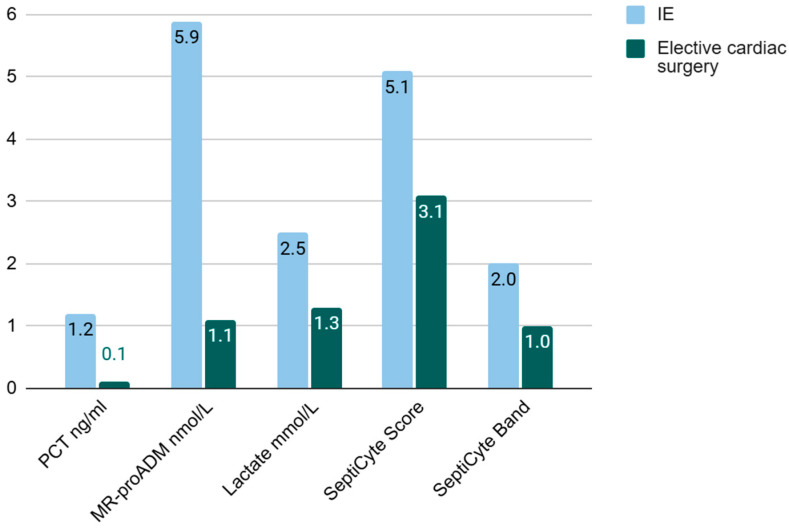
Comparison of median biomarker levels between IE patients and elective cardiac surgery patients. Abbreviations: IE: Infective endocarditis, PCT: procalcitonin, MR-proADM: mid-regional pro-adrenomedullin.

**Table 1 ijms-26-09453-t001:** Characteristics of Group 1—infectious endocarditis.

ID	Sex/ Age	Comorbidities	Charlson Comorbidity Index	ASA Score	Euro Score	SOFA	SAPS	Type of Cardiac Surgery	CPB Duration	Inotropes- Vasopressors	Vasoplegia Score	Isolated Pathogen-Sample Type	Clinical Suspicion	Length of Stay in ICU Days
1	M/53	HT, CKD	4	4	38.6	15	58	valve	287	A, NA	Severe	*Candida albicans*-valve	Septic vs. cardiogenic shock	13
2	M/64	Ex smoker, alcohol-dependent, active neoplastic disease	5	4	4.9	8	48	valve	180	DBT, NA	Moderate	*E. faecalis*- blood	Cardiogenic shock	15
3	F/75	HT, CKD, DM	5	4	23.8	12	52	valve	171	A	Mild	*E. coli*- blood	Cardiogenic shock	10
4	M/76	COPD	5	3	9.2	10	57	valve	134	DBT, NA, VP	Moderate	MSSE- blood	Septic shock	24
5	M/76	HT, previous cancer, diabetes	4	4	28	8	57	valve	150	A, NA	Mild	MSSE- valve		1
6	F/61	HT, COPD, DM	5	4	9.1	12	59	valve	220	A, NA	Moderate	MSSE- blood	Septic vs. cardiogenic shock	2
7	F/78	HT	4	4	19.5	11	63	valve	149	A, NA	Moderate	MRSE- blood	Septic vs. cardiogenic shock	2
**Mean/** **median**	**69/** **75**		**4.6/** **5**	**3.9/** **4**	**19/** **19.5**	**10.9/** **11**	**56.3/** **57**		**184/** **171**					**9.6/** **10**

Abbreviations: ID: Identification, M: Male, F: Female, ASA score: American Society of Anesthesiologists physical status classification, Euro score: European System for Cardiac Operative Risk Evaluation, SOFA: Sequential Organ Failure Assessment, SAPS: Simplified Acute Physiology Score, CBP: Cardiopulmonary bypass, ICU: Intensive Care Unit, HT: Hypertension, DM: Diabetes mellitus, COPD: Chronic obstructive pulmonary disease, CKD: Chronic kidney disease. A: Adrenaline, DBT: Dobutamine, NA: Noradrenaline, VP: Vasopressine. Vasoplegia score: Tsiouris’ score. MSSE: Methicillin-Susceptible *S. epidermidis*, MRSE: Methicillin-Resistant *S. epidermidis*.

**Table 2 ijms-26-09453-t002:** Characteristics of Group 2—elective surgery.

ID	Sex/ Age	Comorbidities	Charlson Comorbidity Index	ASA Score	Euro Score	SOFA	SAPS	Type of Cardiac Surgery	CPB Duration	Inotropes- Vasopressors	Vasoplegia Score	Isolated Pathogen- Sample Type	Clinical Suspicion	Length of Stay in ICU Days
1	M/73	Ex smoker, HT, DM	4	3	1.4	5	24	Bypass	148	DBT	Mild			0
2	M/69	Smoker, HT, DM	5	3	5	3	12	Valve	118		Mild			1
3	M/67	COPD	2	3	1	6	39	Valve	163	DBT, NA, VP	Severe		Septic vs. cardiogenic shock	1
4	M/66	HT	5	4	1.4	3	18	Bypass	217		Mild			1
5	M/18	Smoker	1	2	0.7	3	6	Valve	130		Mild			1
6	M/63	Ex smoker	2	3	3	6	34	Valve	158	NA	Mild		Septic vs. cardiogenic shock	3
7	M/69	Smoker, HT, COPD, CKD, DM	5	4	16.1	17	54	Valve	200	A, NA, VP	Mild	*P. aeruginosa*- TA	Septic vs. cardiogenic shock	16
**Mean/** **median**	**61/** **67**		**3.4/** **4**	**3/** **3**	**4.1/** **1.4**	**6.1/** **5**	**26.7/** **24**		**162/** **158**					**3.3/** **1**

Abbreviations: ID: Identification, M: Male, F: Female, ASA score: American Society of Anesthesiologists physical status classification, Euro score: European System for Cardiac Operative Risk Evaluation, SOFA: Sequential Organ Failure Assessment, SAPS: Simplified Acute Physiology Score, CBP: Cardiopulmonary bypass, ICU: Intensive Care Unit, HT: Hypertension, DM: Diabetes mellitus, COPD: Chronic obstructive pulmonary disease, CKD: Chronic kidney disease. A: Adrenaline, DBT: Dobutamine, NA: Noradrenaline, VP: Vasopressine. Vasoplegia score: Tsiouris’ score. TA: Tracheal aspirate.

**Table 3 ijms-26-09453-t003:** Biomarker levels of Group 1—infectious endocarditis.

ID	PCT ng/mL	MR-proADM nmol/L	Lactate mmol/L	SeptiCyte Score	SeptiCyte Band
1	11.4	9.9	6.7	7.5	4
2	1.6	12	1.4	4.7	1
3	3.9	9.6	2.8	7.9	4
4	1.2	5.9	1.7	5.1	2
5	0.4	1.6	2.5	5.6	2
6	0.7	5	2.2	3.3	1
7	0.1	1.4	3.2	5.1	2
**Mean/** **median**	**2.8/** **1.2**	**6.5/** **5.9**	**2.9/** **2.5**	**5.6/** **5.1**	**2.3/** **2**

Abbreviations: ID: Identification, PCT: procalcitonin, MR-proADM: mid-regional pro-adrenomedullin.

**Table 4 ijms-26-09453-t004:** Biomarker levels of Group 2—elective surgery.

ID	PCT ng/mL	MR-proADM nmol/L	Lactate mmol/L	SeptiCyte Score	SeptiCyte Band
1	0.1	1.1	1.3	3	1
2	0.2	1.2	1.6	2.6	1
3	2.4	1.6	1	3.2	1
4	0.1	0.8	1.1	3.6	1
5	0.1	0.8	1.5	3.1	1
6	0.1	1.1	0.8	2.7	1
7	97	12.2	7.2	9.3	4
**Mean/** **median**	**14.3/** **0.1**	**2.7/** **1.1**	**2.1/** **1.3**	**3.9/** **3.1**	**1.4/** **1**

Abbreviations: ID: Identification, PCT: procalcitonin, MR-proADM: mid-regional pro-adrenomedullin.

## Data Availability

Data are available upon request to authors.

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
