# Peer review of "Early Use of Innovative Biomarkers Such as Mid-Regional Pro-Adrenomedullin and SeptiCyte® RAPID in Post-Cardiac Surgery Patients: Pilot Case Series"

_ijms, 2025, doi:10.3390/ijms26199453_

Round 1
Reviewer 1 Report
Comments and Suggestions for Authors
This is a single-center prospective study which compared the levels of two novel biomarkers associated with sepsis and sepsis-related organ dysfunction in patients undergoing elective cardiac surgery with patients undergoing valve surgery for active infection with bacterial endocarditis. The stated purpose is to better define markers to differentiate between the expected non-infectious inflammatory response generated by cardiopulmonary bypass and patients with early bacterial infection, with the goal of earlier and more targeted therapy for infection after cardiac surgery. They studied biomarker levels within 24 hours of cardiac surgery for 7 patients in each group, and found that the "SeptiCyte RAPID" assay performed well in differentiating between post-bypass inflammatory response and patients with bacterial infection, and that the "MR-proADM" assay (which measures the pre-cursor of a substance associated with endothelial damage and organ dysfunction in sepsis) similarly was significantly higher in patients with actual bacterial infection (including one "crossover" patient in the group intended to serve as controls who developed an unexpected bacterial infection).
The study was well designed and the manuscript is extremely well written. The authors appropriately acknowledge their study's primary limitation in the very small sample size. This will be a significant contribution to existing literature in a rapidly evolving field, which offers the real potential to improve the postoperative management of cardiac surgery patients.
I have several minor comments/points for the authors:
1) In paragraphs 1 and 2 of Results, the full terms for the abbreviations SOFA and SAPS should be used with the first citation of these terms (paragraph 1), not the second one in paragraph 2.
2) It is a but curious that the authors did not choose to exclude the patient in Group 2 (intended to be cardiac surgery patients without known infection) who developed a known bacterial infection, which could obviously skew the study results. It is notable, however, that even when this confounding case was included the results still showed significant value using the two measured biomarkers to assess for early evidence of bacterial infection or sepsis.
3) Recommend for Tables 3 and 4 that the authors include the group identities in the Table titles (not just "Group 1" and "Group 2," but instead "Group 1 - infectious endocarditis" and "Group 2 - elective surgery."
4) There is a typographical error in line 226, which should read "IE" not "EI" (for infectious endocarditis).
Well done!
Author Response
We would like to sincerely thank the reviewers for their thoughtful comments, which have helped us to improve the quality and clarity of our manuscript. Below, we provide a detailed, point-by-point response to each comment. All changes in the revised manuscript are highlighted accordingly.
REVIEWER 1
Comments 1: This is a single-center prospective study which compared the levels of two novel biomarkers associated with sepsis and sepsis-related organ dysfunction in patients undergoing elective cardiac surgery with patients undergoing valve surgery for active infection with bacterial endocarditis. The stated purpose is to better define markers to differentiate between the expected non-infectious inflammatory response generated by cardiopulmonary bypass and patients with early bacterial infection, with the goal of earlier and more targeted therapy for infection after cardiac surgery. They studied biomarker levels within 24 hours of cardiac surgery for 7 patients in each group, and found that the "SeptiCyte RAPID" assay performed well in differentiating between post-bypass inflammatory response and patients with bacterial infection, and that the "MR-proADM" assay (which measures the pre-cursor of a substance associated with endothelial damage and organ dysfunction in sepsis) similarly was significantly higher in patients with actual bacterial infection (including one "crossover" patient in the group intended to serve as controls who developed an unexpected bacterial infection).
The study was well designed and the manuscript is extremely well written. The authors appropriately acknowledge their study's primary limitation in the very small sample size. This will be a significant contribution to existing literature in a rapidly evolving field, which offers the real potential to improve the postoperative management of cardiac surgery patients.
Response 1: We would like to thank Reviewer 1 for the thorough review and the highly positive feedback on our work. We are grateful for the recognition of the novelty of applying the proposed biomarkers, even though the current study was conducted on a relatively small sample, as a preliminary step toward further developments. Below, we provide detailed responses to all the points raised.
Comments 2: I have several minor comments/points for the authors:
- In paragraphs 1 and 2 of Results, the full terms for the abbreviations SOFA and SAPS should be used with the first citation of these terms (paragraph 1), not the second one in paragraph 2
Response 2: We thank the reviewer and we acknowledge this insightful suggestion. We modified the sentence inserting the full terms of SOFA and SAPSII where they are first mentioned.
Comments 3:
- It is a but curious that the authors did not choose to exclude the patient in Group 2 (intended to be cardiac surgery patients without known infection) who developed a known bacterial infection, which could obviously skew the study results. It is notable, however, that even when this confounding case was included the results still showed significant value using the two measured biomarkers to assess for early evidence of bacterial infection or sepsis.
Response 3: We thank the reviewer for this interesting and insightful comment. We included in this analysis one patients in the elective surgery group who then developed a P.aeruginosa infection with septic shock. Notably, the patients was not infected, or at least there were no clinical or laboratory findings suggesting infection, at the moment of enrollment. Precisely because the infection was neither identified nor suspected at the time of enrollment (i.e., prior to entering the operating room), we considered it appropriate to retain the patient in the original group, as if performing an evaluation according to the intention-to-treat principle.
Furthermore, this patient presented higher value of Septicyte score and other standard biomarkers, while the other six patients kept lower values of Septicyte score and never developed infection during ICU stay. This data might confirm the potential role of Septicyte assay for early infection diagnosis.
We added a clarifying statement as it follows:
“At the time of enrollment, however, there were no clinical, biochemical, or radiographic signs suggestive of infection, and for this reason the patient was retained in the original group, consistent with an intention-to-treat approach.”
Comments 4:
- Recommend for Tables 3 and 4 that the authors include the group identities in the Table titles (not just "Group 1" and "Group 2," but instead "Group 1 - infectious endocarditis" and "Group 2 - elective surgery."
Response 4: We thank the reviewer for raising this important point. We modified all labels adding “infective endocarditis” in Group 1 and "elective surgery” in Group 2.
Comments 5:
- There is a typographical error in line 226, which should read "IE" not "EI" (for infectious endocarditis).
Response 5: We thank the reviewer for this comment. We corrected the acronym from “EI” to “IE” in line 226.
Reviewer 2 Report
Comments and Suggestions for Authors
The authors are trying to assess the role of new markers of sepsis in patients who have undergone CPR, but a number of questions arise:
In the section "materials and methods", what does simple cardiac surgery mean? Are there such things?In this case, what is difficult about surgery for infectious endocarditis? In my opinion, there is nothing complicated in connecting an artificial circulatory system and replacing the valve with a mechanical one... Therefore, there is no need for speculative phrases, describe all the operations in detail!
I did not see an assessment of Duke's criteria in all patients.
There is no clear description of the data on blood culture for the pathogen, what recommendations do you follow?Where are the links to the naked recommendations on infectious endocarditis?
Where are the echocardiography data for all patients?
How were the sources of sepsis in patients identified or excluded before and after surgery?
The sample group is too small, and it is also not possible to compare patients with the source of infection against an unknown group.
By and large, the article looks like an advertisement for new biomarkers, without clinical and scientific justification.
Author Response
We would like to sincerely thank the reviewers for their thoughtful comments, which have helped us to improve the quality and clarity of our manuscript. Below, we provide a detailed, point-by-point response to each comment. All changes in the revised manuscript are highlighted accordingly.
REVIEWER 2
We would like to thank Reviewer 2 for the time and effort devoted to evaluating our manuscript. We recognize that some comments were expressed in a way that we initially perceived as less constructive, considering the manner in which they were expressed which may not fully align with the style expected in a scientific review.
However, we fully understand that such observations may reflect the reviewer’s genuine concern regarding the limitations of the study,
As already emphasized throughout the manuscript and in the title, we are aware of the limitation related to the small sample size, and we presented this work explicitly as a pilot study intended to precede further developments. While some remarks were challenging to interpret as aimed at improvement, we have nonetheless made every effort to address them constructively.
In this collaborative spirit, we have carefully considered all points raised and incorporated the reviewer’s suggestions wherever feasible, with the goal of further strengthening our manuscript.
Comments 1: The authors are trying to assess the role of new markers of sepsis in patients who have undergone CPR, but a number of questions arise:
In the section "materials and methods", what does simple cardiac surgery mean? Are there such things?In this case, what is difficult about surgery for infectious endocarditis? In my opinion, there is nothing complicated in connecting an artificial circulatory system and replacing the valve with a mechanical one... Therefore, there is no need for speculative phrases, describe all the operations in detail!
Response 1: We thank the reviewer for raising this point. The aim of this pilot study is to evaluate the feasibility of the use of a genomic assay to discriminate infected or not infected patients after Cardio pulmonary bypass (CBP), especially when vasoplegia with all clinical features of distributive shock develops. The term “uncomplicated” focus on the fact that these patients were scheduled for standard cardiac surgery without known critical infections. We acknowledge that this term could be possibly misleading and we modified with “standard elective cardiac surgery”.
The sentences referring to surgical interventions are now modified as it follows:
“Although limited by the small sample, our preliminary data suggest no biomarker alterations in patients with standard elective cardiac surgery, unlike in those with IE.”
“For this purpose, we decided to assess the response of SeptiCyte® RAPID and MR-proADM in two categories of cardiac surgery patients: those undergoing standard elective cardiac surgery (the type of surgery is reported in table 2), and those undergoing valve replacement due to infective endocarditis with a surgical indication”
“Our observations showed no important biomarkers alteration in patients undergoing standard elective cardiac surgery, while they were both increased in patients with IE requiring CPB surgery for valvular replacement”.
Comments 2: I did not see an assessment of Duke's criteria in all patients.
Response 2: We thank the reviewer for this valuable comment. All patients included in the infective endocarditis group of this pilot study had a confirmed diagnosis of IE, established by a multidisciplinary team comprising cardiologists, infectious disease specialists, cardiothoracic surgeons, and anesthesiologists/critical care physicians. The diagnosis was based on a comprehensive assessment that included clinical evaluation, radiological and imaging criteria (namely transesophageal echocardiography), and blood cultures. We have added a clarifying sentence in the Methods section to better specify the diagnostic workup.
The diagnosis of infective endocarditis was confirmed prior to the surgical decision by a multidisciplinary team of physicians, according to Duke criteria. In particular, in addition to clinical, immunological, vascular and microbiological criteria, all patients underwent transesophageal echocardiographic evaluation.
Comments 3: There is no clear description of the data on blood culture for the pathogen, what recommendations do you follow? Where are the links to the naked recommendations on infectious endocarditis?
Response 3: We thank the reviewer for this important comment. We would like to clarify that blood cultures were always obtained prior to defining the surgical indication. The microbiological results were then integrated with clinical, echocardiographic, and imaging findings to establish the diagnosis of infective endocarditis and to define the cases accordingly.
Regarding the recommendations followed, our diagnostic and management approach was based on the current international guidelines, in particular the European Society of Cardiology (ESC) Guidelines for the management of infective endocarditis (2023). We have now added a reference to these guidelines in the Methods section to make this point clearer.
We did not report data of hemocultures in this paper since it goes beyond our primary aim. According to your comments, however, we specify both the diagnostic criteria both the referral guidelines.
Comments 4: Where are the echocardiography data for all patients?
Response 4: We thank the reviewer for this suggestion. All patients underwent a thorough echocardiographic evaluation as part of the diagnostic workup for infective endocarditis. While detailed echocardiographic findings for each patient are not presented, as they are not essential to the primary objectives of this study, these assessments were performed carefully and contributed to the accurate diagnosis of all cases.
Comments 5: How were the sources of sepsis in patients identified or excluded before and after surgery?
Response 5: We thank the reviewer for this important question. Within the 24 hours prior to surgery, each patient underwent a comprehensive clinical evaluation, complemented by laboratory tests and any additional investigations deemed appropriate by the attending anesthesiologist or the cardiothoracic surgeons. Postoperatively, patients were managed in the cardiac intensive care unit, where they received individualized monitoring. This included routine laboratory assessments and, when indicated by the responsible clinicians, microbiological testing to identify or rule out sources of sepsis.
Considering your point, we added in the methods section this paragraph:
“Within the 24 hours prior to surgery, each patient underwent a comprehensive clinical assessment, including laboratory tests and any additional investigations deemed appropriate by the attending anesthesiologist or cardiothoracic surgeons. Postoperatively, patients were managed in the cardiac intensive care unit with individualized monitoring, encompassing routine laboratory tests and microbiological analyses when indicated by the responsible clinicians, to identify or rule out potential sources of sepsis.”
Comments 6: The sample group is too small, and it is also not possible to compare patients with the source of infection against an unknown group.
Response 6: We fully agree with the reviewer regarding the small sample size, which we have emphasized throughout the manuscript as an important limitation. However, given the innovative nature of the study, we aimed to highlight the observations obtained from this initial patient cohort as a pilot study. In addition, no comparative analysis between patients with a known source of infection and those without was performed; rather, the study provides a preliminary descriptive assessment intended to illustrate the potential utility of SeptiCyte and pro-ADM in this context.
Comments 7: By and large, the article looks like an advertisement for new biomarkers, without clinical and scientific justification.
Response 7: We thank the reviewer for this comment. We respectfully note that all statements regarding the biomarkers in our manuscript are supported by relevant literature and directly linked to the data presented. We have made every effort to ensure that the discussion remains evidence-based and clinically justified, without overstating the significance of our findings.
Round 2
Reviewer 2 Report
Comments and Suggestions for Authors
Unfortunately, the authors were unable to improve the manuscript, and I still adhere to my original position.